# The Prognostic Significance of FOXD1 Expression in Head and Neck Squamous Cell Carcinoma

**DOI:** 10.3390/jpm13030530

**Published:** 2023-03-15

**Authors:** Wenmei Jiang, Yudong Li, Ruiyu Li, Wenkuan Chen, Ming Song, Quan Zhang, Shuwei Chen

**Affiliations:** Department of Head and Neck Surgery, Sun Yat-sen University Cancer Center, State Key Laboratory of Oncology in South China, Collaborative Innovation Center for Cancer Medicine, Guangzhou 510060, Chinasongming@sysucc.org.cn (M.S.)

**Keywords:** forkhead box D1, head and neck squamous cell carcinoma, clinicopathologic characteristics, survival

## Abstract

It has been reported that forkhead box D1 (FOXD1) plays an established role in human early embryonic development and is broadly involved in various malignancies. However, there is limited information regarding FOXD1 expression in head and neck squamous cell carcinoma (HNSCC). This present study aimed to explore the clinical significance of FOXD1 in patients with HNSCC. Tissue microarrays of 334 primary HNSCC patients who underwent surgery between 2008 and 2010 at Sun Yat-sen University Cancer Center were investigated by immunohistochemistry regarding FOXD1 expression. χ^2^ test was used to estimate the relationship of FOXD1 expression with clinicopathologic characteristics. Univariate and multivariate analyses were performed to identify FOXD1 expression as an independent prognostic indicator of overall survival (OS) and disease-free survival (DFS). FOXD1 expression is closely associated with postoperative recurrence. HNSCC patients with high FOXD1 expression have poorer prognoses than the low-expression group (*p* < 0.05). According to multivariate analysis, FOXD1 was an independent prognostic factor for OS and DFS. The results revealed that FOXD1 could be a prognostic factor for HNSCC and might serve as a potential target for novel therapies.

## 1. Introduction

Head and neck cancers are the seventh most common malignancy in the world with 977,171 new cases and 461,774 deaths annually [1]. The anatomic subsites of the head and neck malignant tumors include naso-, oro- and hypopharynx, larynx, nasal cavity, oral cavity, the floor of the mouth, the palate, tongue, tonsils, salivary glands, etc. [2]. Squamous cell carcinoma (SCC) accounts for approximately 90% of head and neck malignancies [3]. Among them, the oral cavity SCC (OCSCC) is the most common malignancy of head and neck SCC (HNSCC), which is the first leading cause of cancer death in Southern Asia [4,5]. In China, there are 52,200 new cases and 25,800 deaths of head and neck carcinoma every year, with a male–female ratio of over 2 [6]. HNSCC has a high potential for local invasion and node metastases, and about 70% of advanced cases are incurable [7]. Due to the unsatisfied awareness of HNSCC, a large part of patients present with advanced disease at diagnosis, which accounts for the poor prognosis and high recurrence of HNSCC. The overall 5-year survival rates of HNSCC consistently remain around 50% and have not improved over the past 30 years [4]. Tobacco and alcohol are the established causes of HNSCC; however, its tumorigenesis is complicated and involves many crucial genes and signaling pathways [8,9]. Therefore, it is meaningful to explore the underlying molecular biology mechanisms of the pathogenesis of HNSCC and to identify novel drug targets for the disease.

Forkhead box D1 (FOXD1), one of the forkhead box family, functions as a transcription factor cell cycle [10]. Transcription factors that contain a forkhead domain play an essential role in kidney morphogenesis and in specification of the temporal retina in mammals [11,12]. The FOXD1 protein also has a role in a wide array of biological processes, including proliferation, metabolism control and tumorigenesis [13,14]. FOXD1 is upregulated in prostate cancer, lung cancer and breast cancer [15,16,17]. Conversely, FOXD1 is reported to suppress the proliferation of ovarian carcinoma cells and is correlated with chemoresistance of ovarian cancer patients [18]. While previous studies suggest that FOXD1 is broadly involved in various malignancies, there is limited information regarding its significance in OCSCC [19].

In the present study, tissue microarrays (TMAs) were used to estimate FOXD1 expression via immunohistochemical staining (IHC) based on the tissue samples of 334 HNSCC patients. Herein, it was indicated for the first time that FOXD1 is significantly related to the poor prognosis of patients with HNSCC and is a prognostic marker for HNSCC.

## 2. Materials and Methods

### 2.1. Subjects and Samples

The study was approved by the Clinical Research Ethic Committee of Sun Yat-sen University Cancer Center (approval number: B2022-221-01), and the informed consent of patients was waived. This retrospective study included patients with HNSCC who underwent surgery between 2008 and 2010 in the Department of Head and Neck Surgery, Sun Yat-sen University Cancer Center, Guangzhou, China. The enrolled cases reached all the following criteria: (1) patients were histologically diagnosed as SCC of the head and neck for the primary malignancy; (2) neither chemotherapy nor radiotherapy was administered prior to surgery; (3) the absence of distant metastases; (4) adequate clinical information and constant follow-up. Excluding 5 cases of tissue detachment and 29 cases of invalid immunohistochemical staining, eventually, 334 eligible HNSCC cases were enrolled in our study, with 282 males and 52 females, and their paraffin-embedded samples were collected after surgery. Patient characteristics included age at diagnosis, gender, histological grade, tumor subsites, smoking/drinking status, American Joint Committee on Cancer (AJCC) stage, T stage, N stage, and treatment modality (surgery alone or surgery with adjuvant radiotherapy and/or chemotherapy). All 334 HNSCC patients were constantly followed up by telephone, e-mail or at an outpatient review. All patients had agreed with the collection of tissue samples and already signed the informed consent. This study was approved by the Medical Ethics Committee of the Sun Yat-sen University Cancer Center.

### 2.2. Tissue Microarray (TMA)

Two experienced pathologists independently reviewed the histology of all cases used in this study. Hematoxylin and eosin (HE) slides of the paraffin-embedded tissue blocks were carefully reviewed to mark the most representative areas and rough extension of the tumor. TMAs were constructed according to the establish protocol reported earlier [20]. Tissue cores were then extracted from the marked paraffin-embedded blocks via the tissue-arraying instrument (Beecher Instruments) and transferred into recipient blocks. TMA slides were produced (thickness 2–3 μm) to generate slides for immunochemistry (IHC) staining.

### 2.3. Immunohistochemistry

The TMAs slides were firstly deparaffinized and then incubated in xylene, followed by different concentrations of ethanol for rehydration, and then washed with distilled water. For endogenous peroxidase ablation, the slides were immersed into a Coplin jar containing 3% hydrogen peroxide for 15 min. For epitope retrieval, the slides immersed with sodium citrate buffer (pH 6.0) were boiled in a Pascal Pressurized Heating Chamber for 10 min. After rinsing with phosphate-buffered saline (PBS) buffer, the slides were blocked with normal goat serum. Then, the slides were incubated overnight with FOXD1 antibody (1:50; Thermo Fisher Scientific, catalog # PA5-35145, RRID AB_2552455) in a humid chamber at 4 °C. After PBS washes, a peroxidase-labeled secondary antibody (EnVision/HRP system, DAKO, Carpinteria, CA, USA) was applied to incubate the slides for 1 h, and then, the slides were stained with 3,3′-diaminobenzidine tetrahydrochloride. Finally, these slides were stained with hematoxylin. After differentiation, dehydration, and coverslip, the slides were observed under a microscope. 

### 2.4. Evaluation of FOXD1 Expression

The expression of FOXD1 was evaluated by a semi-quantitative method [21,22], and this evaluation was carried out by two independent pathologists in a blinded fashion. The intensity of immunostaining was calculated as points 0 for no staining, 1 for very weak staining (light yellow), 2 for moderate staining (yellow brown) and 3 for strong staining (brown). The proportion of immunoreactive tumor cells scoring was evaluated as follows: 0, no positive cells; 1, <10% positive cells; 2, 10–50% positive cells; 3, 51–80% positive cells; 4, >80% positive cells. The immunoreactivity score was calculated by the intensity of immunostaining multiplied by the proportion of immunoreactive tumor cells. The median immunoreactivity scores from two independent pathologists were used as the final scores, ranging from 0 to 12. Accordingly, the patients were divided into FOXD1 high-expressed group (immunoreactivity scores ≥2) and low-expressed group (immunoreactivity scores <2) dependent on the median value of immunoreactivity scores, which is 2.

### 2.5. Statistical Analyses

All statistical analyses were performed using SPSS Statistics 25.0 software (IBM SPSS, Inc., Chicago, IL, USA), and R version 3.5.2 (https://www.r-project.org/ (accessed on 21 July 2021)). Correlations between FOXD1 expression and patients’ clinicopathological characteristics mentioned above were analyzed by χ^2^ test. Univariate and multivariate analyses were performed based on Cox proportional hazard regression model to identify if the expression of FOXD1 was an independent prognostic factor of HNSCC. Hazard ratios (HR) with 95% confidence intervals (CIs) were calculated using multivariate regression analysis. Kaplan–Meier analysis and log-rank tests were applied to estimate and compare survival curves between the high-expression and low-expression groups, that is overall survival (OS) and disease-free survival (DFS). A two-sided *p* < 0.05 was considered of statistical significance in the study.

### 2.6. Protein-Protein Interaction of FOXD1

We used a database available online (Https://string-db.org/ (accessed on 21 July 2021)) to analyze the protein interaction network related to FOXD1. 

## 3. Results

### 3.1. Patients Characteristics

Among the 334 patients with HNSCC enrolled in our study (Figure 1), the majority were males (282, 84.4%), and the median age was 55 years (range, 28–90 years). The median follow-up was 62.1 months (range, 5–131 months). The TNM stage at diagnosis was advanced (AJCC TNM stage III and IV) in 206 patients (61.7%) and early (AJCC TNM stage I and II) in 128 patients (38.3%). According to the pathological stage, over half of the patients presented (194, 58.1%) with T1 or T2 stage at their first diagnosis, and 155 patients (46.4%) presented with N+ stage. The majority of tumors turned out with well or moderate histological grade. A total of 195 patients (58.4%) with HNSCC underwent surgery alone, and the rest (135, 41.6%) underwent surgery with adjuvant radiotherapy and/or chemotherapy. More than half of the enrolled patients had a history of smoking (196, 58.7%). During our long-term follow-up, 79 of the enrolled patients suffered recurrence or metastases. There were 167 deaths until our latest follow-up, and among them, 143 deaths were attributed to tumor. The demographic and clinicopathologic characteristics are listed in Table 1.

### 3.2. Immunostaining of FOXD1

The representative FOXD1 immunostaining of HNSCC tumor samples are demonstrated in Figure 2. FOXD1 is mainly expressed in the cytoplasm. Of the entire HNSCC tumor samples, 199 cases (59.6%) presented with a high expression level of FOXD1. The FOXD1 expression level showed no obvious relationship with AJCC stage, T stage, N stage, age, sex, smoking/drinking status, or the treatment strategy (all *p* > 0.05). Furthermore, no statistical significance was found between FOXD1 expression and subsites or histological grades (overall *p* > 0.283, Figure 3A,B). However, HNSCC patients with a high level of FOXD1 expression tended to present with relapse or metastases status during the follow-up versus those with a low expression level of FOXD1 (*p* = 0.021, Figure 3C). The expression level of FOXD1 had a statistical relationship with death events (*p* = 0.029, Figure 3D). Table 1 presents the association between FOXD1 expression levels and clinicopathological characteristics of HNSCC patients. 

### 3.3. Correlation between Survival and FOXD1 Expression

Univariate analysis revealed that gender, histological grade, subsites, T stage, N stage AJCC stage, treatment modality, and FOXD1 expression level were associated with the overall survival of HNSCC patients (all *p* < 0.05, Table 2). However, T stage and subsites had no statistical relationship with DFS (all *p* > 0.05). 

The multivariate analysis based on the Cox hazards regression model of OS showed that FOXD1 expression was estimated to be among the most significant independent prognostic factors for HNSCC patients, along with AJCC stage (all *p* < 0.05, Table 2). FOXD1 expression is the only independent prognostic factor of DFS (*p* = 0.006). Kaplan–Meier analysis of FOXD1 expressions was performed, and the survival curves for HNSCC patients are shown in Figure 4. Notably, survivals of HNSCC patients were statistically associated with FOXD1 expression. Patients with high expression of FOXD1 had a worse prognosis than cases with low FOXD1 expression level (*p* < 0.05). The 5-year OS among the FOXD1 high-expression group was much lower than the low-expression group (*p* = 0.0078, Figure 4A), as was the 5-year DFS (*p* = 0.0078, Figure 4B). 

### 3.4. Protein-Protein Interaction of FOXD1

We used a database available online (Https://string-db.org/ (accessed on 21 Jul 2021)) to analyze the protein interaction network related to FOXD1 as shown in Figure 5. FOXD1 might be co-expressed with SIX1 and SIX2 of the SIX family, as well as PAX2, EYA1, WT1, and CITED1. In addition, FOXD1 had a close relationship with SALL1. The relationship between the interacted genes and overall survival of patients derived from TCGA is shown in Appendix A Figure A1. Among these interacted genes, the expression of WT1 had a statistical relationship with the overall survival of patients with head and neck carcinoma in the TCGA database (*p* = 0.019). In this figure, SIX1 represents the SIX family, and there were no data regarding PAX2 and SALL1 in head and neck carcinoma according to the TCGA database.

## 4. Discussion

It is well established that the FOX family, encoding transcription factors with forkhead motifs, are broadly involved in regulating physical and biological processes, including embryonic development and organogenesis, cell cycle regulation, and tumorigenesis. The dysfunction of FOXD1 has been reported to be associated with several malignancies [10,11,12,13,14]. To explore the role of FOXD1 expression in HNSCC and to validate if there is a relationship between FOXD1 expression and the prognostic situation of patients with HNSCC, this present study performed immunohistochemical staining on the TMAs of 334 HNSCC patients. In our study, FOXD1 expression is the independent prognostic factor of OS and DFS. Patients with a higher level of FOXD1 expression tend to present with lower 5-year overall survival, indicating a relatively poorer prognosis compared with those patients with lower FOXD1 expression.

Several previous studies have tried to elucidate the role of FOXD1 involved in tumorigenesis and progression [23,24,25]. It has been reported that in normal prostate tissue, FOXD1 was hardly expressed; however, it was highly expressed in prostate cancer cells and lymph node metastases, indicating the contribution of FOXD1 in prostate tumorigenesis [15]. A microarray analysis assessed the levels of FOXD1 mRNA in non-small cell lung cancer (NSCLC) and found that the levels were significantly associated with patient survival by disturbing cell proliferation [17]. NSCLC patients with higher FOXD1 expression have significantly lower survival rates than those with normal FOXD1 levels, indicating that the expression of FOXD1 might induce novel treatment strategies for NSCLC [26]. FOXD1 might also be related to chemotherapeutic drug resistance for patients with breast cancer by targeting p27 expression and then inducing G1 to S phase transition in the cell cycle [27]. However, little is known about the role of FOXD1 in HNSCC.

We found by immunostaining that high-level expression of FOXD1 was associated with the comparatively poorer prognosis in patients with HNSCC. This phenomenon is consistent with previous studies. Huang et al. and Li et al. found that FOXD1 was highly expressed in carcinoma tissues, and this high expression turned out to be the independent prognostic factors that had a significant relationship with the survival of patients [28,29]. Different from what Li et al. had demonstrated [30], we noted that the expression of FOXD1 was not statistically related to advanced TNM stage and N+ status, but a high expression of FOXD1 was correlated with recurrence or metastasis. Previous studies had revealed that overexpression of FOXD1 promoted the epithelial–mesenchymal transition, which might be the potential mechanism for the poorer survival that presented among patients with a higher FOXD1 expression level [31]. It was reported that high expression of FOXD1 was correlated with pathological differentiation in colorectal carcinoma; however, there was no statistical relationship between histological grade and FOXD1 expression in our study about HNSCC [32]. Although some studies have tried to elucidate the pathways and relating molecular factors including FOXD1 in malignancies [23,25], the mechanism about how FOXD1 regulates cell proliferation in HNSCC remains unclear. Tumor metastases are responsible for most deaths in patients with malignancies, and it is a well-accepted fact that tumor angiogenesis significantly contributes to the process of tumor metastasis [27]. 

According to the protein–protein interactions reached from the database online, FOXD1 might be co-expressed with WT1, SIX1 and SIX2 of the SIX family, PAX2, EYA1, SALL1 and CITED1, which was demonstrated in previous studies that FOXD1 is involved in kidney development along with the genes mentioned above [33,34,35,36]. By reviewing the literature, WT1 drew our attention, and based on the TCGA database, WT1 is the only one among the interacted genes that had a statistical relationship with the survival of patients with head and neck carcinoma. It had been reported that WT1 functions as an oncogene in non-small cell lung cancer and breast cancer [37,38,39]. In the study of Yusuke Oji et al., the WT1 gene demonstrated to be overexpressed at both the mRNA and protein levels in HNSCC cells, and high WT1 expression levels statistically correlated with a high N stage in patients with HNSCC [40]. Xingru Li et al. reported the oncogenic roles of WT1 and p63 in HNSCC cells and proved that WT1 was able to directly regulate p63 expression and induce an effect on several known p63 target genes [41]. WT1 has been reported to be associated with FOXD1 in the development of kidney, and FOXD1 acted as the upstream regulator of WT1 [33]. Whether or not WT1 and FOXD1 have any relationship in tumorigenesis has still not been explored yet. Therefore, to explore the potential mechanism of how FOXD1 affects the activities of HNSCC cells, further study should be arranged.

This study showed a positive relationship between FOXD1 expression and patient survival. However, some limitations should be taken into consideration. The predominant limitation of our study was that it was a single-center, retrospective study with a relatively small cohorts of patients. There exists a need to confirm the results through multicenter studies with larger cohorts. 

In summary, FOXD1 was estimated by IHC on TMAs to present with clinical significance in patients of HNSCC. The high FOXD1 expression correlated with poor prognosis of HNSCC; therefore, FOXD1 expression was identified as an independent prognostic factor of HNSCC. Although the exact mechanism for how FOXD1 expression is regulated remains unclear, based on the results of this present study, we might make a hypothesis that FOXD1 expression could be a prognostic factor for HNSCC and might serve as a potential target for novel therapies.

## Figures and Tables

**Figure 1 jpm-13-00530-f001:**
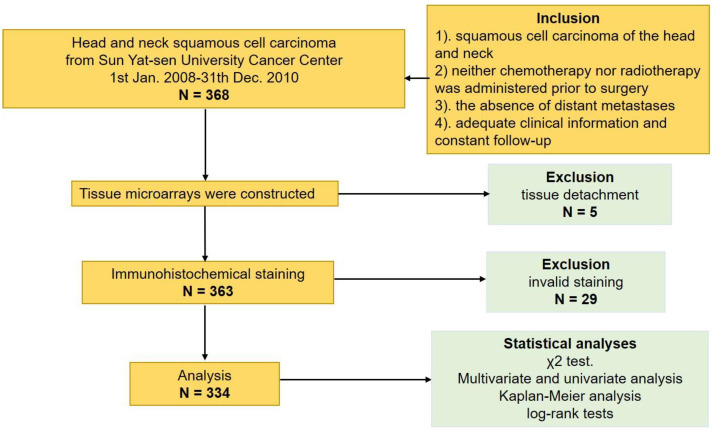
The flow diagram of this study.

**Figure 2 jpm-13-00530-f002:**
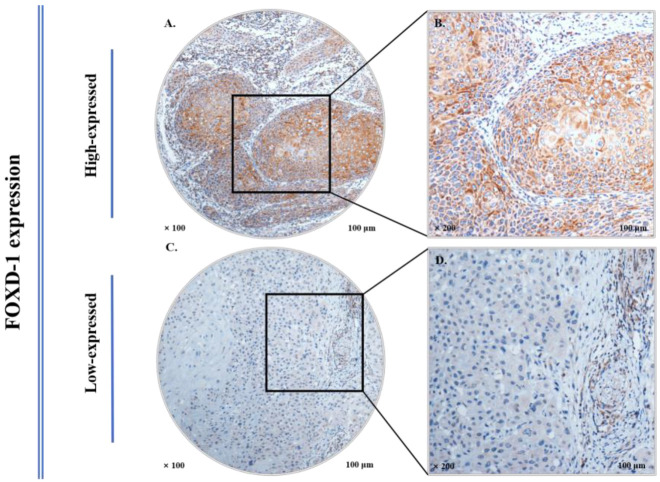
Representative immunohistochemical staining of FOXD1 in HNSCC TMAs. High expression of FOXD1 in 100× (**A**) and 200× (**B**). Low expression of FOXD1 in 100× (**C**) and 200× (**D**).

**Figure 3 jpm-13-00530-f003:**
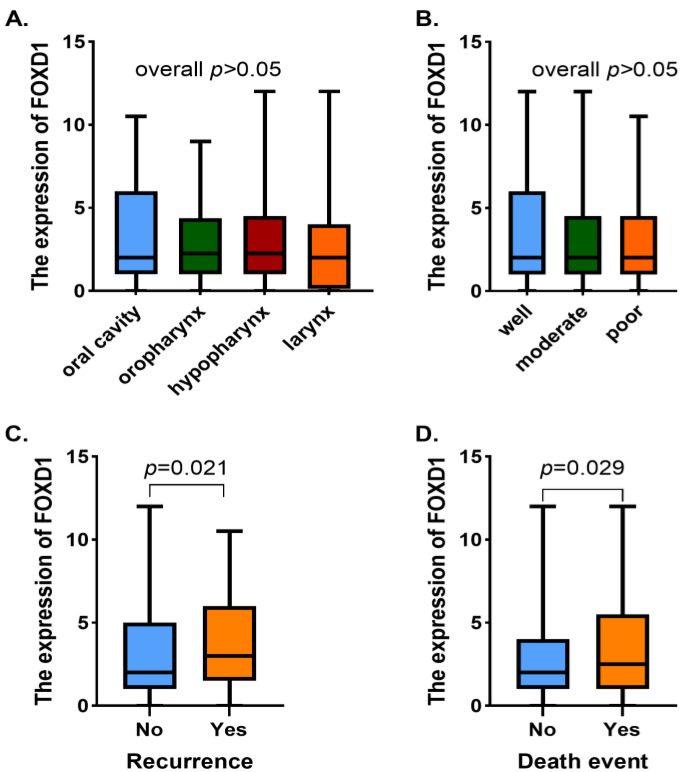
Association between subsites (**A**), histological grade (**B**), recurrence/metastasis (**C**) or death event (**D**) and FOXD1 expression.

**Figure 4 jpm-13-00530-f004:**
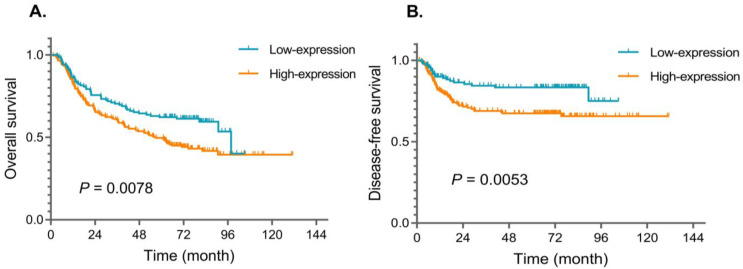
Overall survival (**A**) and disease-specific survival (**B**) curves for cohort of patients with stage I-IV HNSCC according to the expression level of FOXD1.

**Figure 5 jpm-13-00530-f005:**
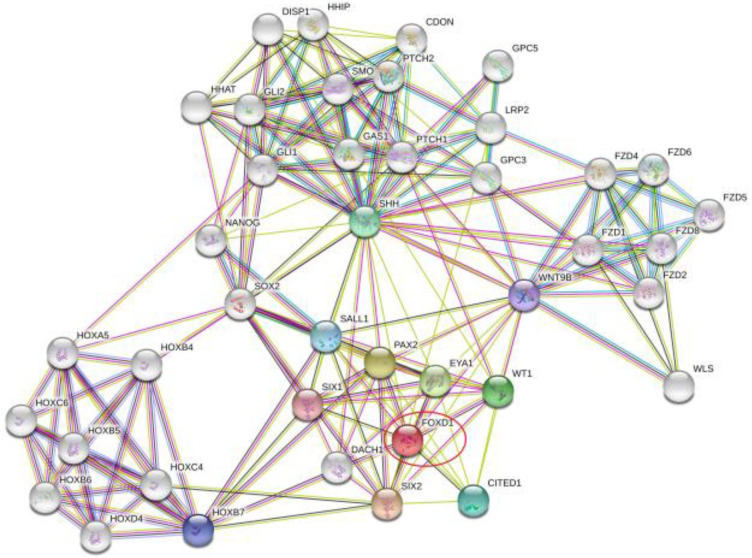
The prediction regarding protein–protein interactions of FOXD1 is shown according to the database available online (https://string-db.org/ (accessed on 21 July 2021)). FOXD1 might be co-expressed with SIX1 and SIX2 of the SIX family, as well as with PAX2, EYA1, WT1, and CITED1. In addition, FOXD1 has a close relationship with SALL1.

**Table 1 jpm-13-00530-t001:** Associations between FOXD1 expression level and clinicopathological characteristics of HNSCC patients.

Variables	Number of Cases	FOXD1 Expression		Kappa Value	*p* Value
		Low N (%)	High N (%)		
**Total**	334	135	199		
**Sex**				2.382	0.123
Male	282	119 (88.1)	163 (81.9)		
Female	52	16 (11.9)	36 (18.1)		
**Age (year)**				0.560	0.454
≤55	128	55 (40.7)	73 (36.7)		
>55	206	80 (59.3)	126 (63.3)		
**Histology grade**				0.763	0.683
Well	130	54 (40.0)	76 (38.2)		
Moderate	124	52 (38.5)	72 (36.2)		
Poor	80	29 (21.5)	51 (25.6)		
**Subsite**				0.410	0.938
Oral cavity	146	61 (45.2)	85 (42.7)		
Oropharynx	32	12 (8.9)	20 (10.1)		
Hypopharynx	80	33 (24.4)	47 (23.6)		
Larynx	76	29 (21.5)	47 (23.6)		
**Smoking status**				1.171	0.279
Non-smokers	138	51 (37.8)	87 (43.7)		
Ex-smokers	196	84 (62.2)	112 (56.3)		
**Drinking status**				0.235	0.628
Non-drinkers	235	93 (68.9)	142 (71.4)		
Ex-drinkers	99	42 (31.1)	57 (28.6)		
**T stage**				2.939	0.086
T1–T2	194	86 (63.7)	108 (54.3)		
T3–T4	140	49 (36.3)	91 (45.7)		
**N stage**				1.596	0.207
N0	179	78 (57.8)	101 (50.8)		
N+	155	57 (42.2)	98 (49.2)		
**AJCC stage**				2.775	0.096
I–II	128	59 (43.7)	69 (34.7)		
III–IV	206	76 (56.3)	130 (65.3)		
**Treatment**				0.034	0.853
Surgery alone	195	78 (57.8)	117 (58.8)		
Surgery with adjuvant radiotherapy and/or chemotherapy	139	57 (42.2)	82 (41.2)		
**Relapse/Metastases**				8.227	0.004
No	255	114 (84.4)	164 (71.9)		
Yes	79	21 (15.6)	64 (28.1)		

**Table 2 jpm-13-00530-t002:** Univariate and multivariate analysis on overall survival and disease-free survival by Cox regression model.

Variables	Univariate		Multivariate	
	HR (95% CI)	*p* Value	HR (95% CI)	*p* Value
** *OS* **				
**Sex**				
Male	1 (ref)		1 (ref)	
Female	0.520 (0.310, 0.871)	0.013	0.764 (0.431, 1.354)	0.357
**Age (year)**				
≤55	1 (ref)			
>55	1.225 (0.891, 1.685)	0.212		
**Histology grade**				
Well	1 (ref)		1 (ref)	
Moderate	1.323 (0.919, 1.903)	0.132	0.940 (0.628, 1.407)	0.764
Poor	1.754 (1.193, 2.580)	0.004	0.918 (0.578, 1.459)	0.719
**Subsites**				
Oral cavity	1 (ref)		1 (ref)	
Oropharynx	1.157 (0.645, 2.076)	0.624	1.334 (0.711, 2.505)	0.369
Hypopharynx	2.033 (1.045, 2.939)	0.000	1.181 (0.756, 1.845)	0.465
Larynx	1.432 (0.954, 2.150)	0.083	1.179 (0.738, 1.885)	0.490
**Smoking status**				
Non-smokers	1 (ref)			
Ex-smokers	1.262 (0.922, 1.728)	0.146		
**Drinking status**				
Non-drinkers	1 (ref)			
Ex-drinkers	1.321 (0.958, 1.820)	0.094		
**T stage**				
T1–T2	1 (ref)		1 (ref)	
T3–T4	2.383 (1.752, 3.243)	0.000	1.229 (0.829, 1.821)	0.304
**N stage**				
N0	1 (ref)		1 (ref)	
N1	2.579 (1.884, 3.531)	<0.001	1.398 (0.883, 2.214)	0.153
**AJCC stage**				
I–II	1 (ref)		1 (ref)	
III–IV	4.037 (2.728, 5.974)	<0.001	2.746 (1.518, 4.969)	0.001
**Treatment**				
Surgery alone	1 (ref)		1 (ref)	
Surgery with adjuvant radiotherapy and/or chemotherapy	1.622 (1.197, 2.197)	0.002	0.913 (0.639, 1.302)	0.614
**FOXD1 expression**				
Low expression	1 (ref)		1 (ref)	
High expression	1.545 (1.119, 2.135)	0.008	1.403 (1.016, 2.129)	0.042
** *DFS* **				
**Sex**				
Male	1 (ref)		1 (ref)	
Female	0.503 (0.300, 0.842)	0.009	0.741 (0.359, 1.530)	0.417
**Age (year)**				
≤55	1 (ref)			
>55	0.705 (0.453, 1.097)	0.124		
**Histology grade**				
Well	1 (ref)			
Moderate	0.774 (0.467, 1.281)	0.319		
Poor	0.864 (0.485, 1.539)	0.620		
**Subsite**				
Oral cavity	1 (ref)			
Oropharynx	0.587 (0.231, 1.491)	0.263		
Hypopharynx	0.637 (0.340, 1.195)	0.160		
Larynx	1.209 (0.716, 2.039)	0.478		
**Smoking status**				
Non-smokers	1 (ref)			
Ex-smokers	0.970 (0.621, 1.514)	0.893		
**Drinking status**				
Non-drinkers	1 (ref)			
Ex-drinkers	0.977 (0.597, 1.599)	0.926		
**T stage**				
T1–T2	1 (ref)			
T3–T4	1.399 (0.897, 2.182)	0.139		
**N stage**				
N0	1 (ref)		1 (ref)	
N1	1.665 (1.068, 2.596)	0.024	1.117 (0.599, 2.084)	0.727
**AJCC stage**				
I–II	1 (ref)		1 (ref)	
III–IV	1.849 (1.142, 2.993)	0.010	1.416 (0.736, 2.726)	0.298
**Treatment**				
Surgery alone	1 (ref)		1 (ref)	
Surgery with adjuvant radiotherapy and/or chemotherapy	1.626 (1.045, 2.529)	0.031	1.311 (0.789, 2.179)	0.296
**FOXD1 expression**				
Low expression	1 (ref)		1 (ref)	
High expression	2.041 (1.238, 3.363)	0.005	2.017 (1.222, 3.330)	0.006

Abbreviations: HR, hazard ratio; CI, confidence interval; OS, overall survival; DFS, disease-free survival; FOXD1, Forkhead box D1; AJCC, American Joint Committee on Cancer.

## Data Availability

Any researchers interested in this study could contact us for requiring the data.

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
