# Peer review of "The Prognostic Significance of FOXD1 Expression in Head and Neck Squamous Cell Carcinoma"

_jpm, 2023, doi:10.3390/jpm13030530_

Round 1

Reviewer 1 Report

More expanded discussion and additional details regarding some of the gene- proteins identified in the online database would add more depth to this paper. To simply indicate that FOXD1 has potential associations with 6 protein and relate them back to other cancers is inadequate. Additional discussion of what these proteins may do and how they impact HNSCC would be relevant even if to do a TCGA analysis to again assess impact on survival, DFS or loco-regional control.

Author Response

Dear Reviewer 1,

Sincere thanks for your professional advice. Here are our responses and all the corrections were marked in red in the revised manuscript.

In the fourth paragraph of the discussion part, we focused on the interacted genes of FOXD1. By reviewing the literature, we found that there were some articles reported the relationship between WT1 and head and neck carcinoma, but nearly no research about the other genes in head and neck carcinoma was reported. Therefore, we mainly concentrated on the potential function of WT1 in head and neck carcinoma. And this might be the basis of our further research in the future. In addition, we explored the relationship between the interacted genes and overall survival of patients with head and neck carcinoma in TCGA, and the results were added as Appendix Figure A1 in the revised manuscript.

Kind regards, 

Wenmei Jiang

Reviewer 2 Report

Jiang et al has observed the prognostic significance of FOXD1 in oral squamous cell carcinoma in the present study. Overall, the study is well presented, conducted, and statistical analysis supports their finding. There are few points that needs to be addressed in this study.

1.       There are several studies that explored the prognostic importance of and expression of FOXD1 and oral squamous cell carcinoma in depth (doi: 10.1111/odi.14002 ;  doi: 10.1186/s13578-021-00671-9; https://doi.org/10.1042/BSR20210158 etc. ) the authors need to discuss about the novelty of their study, or if the findings are similar to the existing literature.

 2.       Section 3.3: this section seems to be incomplete. I do not find any relevance of this section with the present study. Did the authors perform any functional/ observational studies with the co-expressing proteins?

 3.       Line 201-203 : I did not understand of including these lines was that a typing error ?

Author Response

Dear Reviewer 2,

Sincere thanks for your professional review. Here are our responses and all the corrections were marked in red in the revised manuscript.

  1. There are several studies that explored the prognostic importance of and expression of FOXD1 and oral squamous cell carcinoma in depth (doi: 10.1111/odi.14002 ;  doi: 10.1186/s13578-021-00671-9; https://doi.org/10.1042/BSR20210158 etc. ) the authors need to discuss about the novelty of their study, or if the findings are similar to the existing literature.

Response 1: Thank you for your significant advice. We have added some discussion in the third paragraph of the discussion part which was marked in red. We carried out the clinical study based on the tissue microarrays from our center, and our results were consistent with that of the published clinical studies about head and neck carcinoma. We hope these findings derived from a single cancer center could be the solid research basis for further study in head and neck carcinoma.

  1.  Section 3.3: this section seems to be incomplete. I do not find any relevance of this section with the present study. Did the authors perform any functional/ observational studies with the co-expressing proteins.

Response 2: Thank you for the reminder. We have added the corresponding method part as Section 2.6. We feel sorry that no functional studies has been carried out to further verify the relationship. And we added the results about the relationship between these interacted proteins and the survival of patients with head and neck carcinoma based on the TCGA (Appendix Figure A1).  

       3. Line 201-203 : I did not understand of including these lines was that a typing error ?

Response 3: Sincere sorry for our carelessness. The lines 201 to 203 were deleted for they are the remnant of the template. Thank you for your careful observation.

Reviewer 3 Report

It is an interesting manuscript.

My observation are:

Specify if “peripheral regions” page 2 line 82, refers to invasive front or to which region?

In the immunohistochemistry technique (2.3. Immunohistochemistry), because not placing any reference on which based their methodology, it is necessary to specify if:

1. Some buffer was used for the recovery process in the Pascal Pressurized Heating Chamber

2. Indicate which buffer was used after incubation with hydrogen peroxide (not peroxidase), as well as the incubation time for both.

3. It is interesting that no buffer or compound was used for non-specific background inhibition

4. The concentration, time and incubation temperature for the primary antibody must be mentioned

5. What was the nuclear counterstaining method?

6. With what medium or sealing protocol for coverslide was employed?

In the section of “2.4. Evaluation of FOXD1 expression” must be specified

1. Most of the scores are obtained by sum of intensity and proportion, however here it is obtained from multiplication, it necessary explain/justify why it was obtained like this

2. Apparently the scale for the immunoreactivity score is underestimated; the range is from 0 to 12, and it is called high expression from 2, what is the justification?

The result 3.3. Protein-protein interaction of FOXD1 is not presented in the Materials and Methods, the method of how this interaction network was obtained must be included, additionally in this section in the lines 201 to 203 are not comprehensible, it seems that it is a remnant of the template.

It is interesting that there is no statistical relationship of FOXD1 immunoexpression with the degree of differentiation, but yes with more advanced TNM stages and N+, this finding should be considered in the discussion; as well as a more abundant analysis of the relationship with DFS and DSS.

Round 2

Reviewer 3 Report

The manuscript improved significantly, the addition of figure 5 and the appendix figure 1a, allows a better explanation of the correlation finding.

Author Response

Dear reviewer 3,

Sincere thanks for your thorough review. Corrections had made according to your advice, which was marked in red.

  1. The manuscript improved significantly, the addition of figure 5 and the appendix figure 1a, allows a better explanation of the correlation finding.

Response 1: After adding more data about head and neck squamous cell carcinoma, the Figure 5 was deleted, and the content of it was integrated into Figure 2. We also added discussion about it in the third paragraph of the discussion part. As for the appendix figure 1a, discussion was added in the fourth paragraph in the discussion part.

Thank you again for your work.

Kind regards

Wenmei Jiang